# DIVERSITY OF THOUGHT IMPROVES REASONING ABILITIES OF LARGE LANGUAGE MODELS

## ABSTRACT

Large language models (LLMs) are documented to struggle in settings that require complex reasoning. Nevertheless, instructing the model to break down the problem into smaller reasoning steps (Wei et al., 2022), or ensembling various generations through modifying decoding steps (Wang et al., 2023) boosts performance. Current methods assume that the input prompt is *fixed* and expect the decoding strategies to introduce the diversity needed for ensembling. In this work, we relax this assumption and discuss how one can create and leverage variations of the input prompt as a means to *diversity of thought* to improve model performance. We propose a method that automatically improves prompt diversity by soliciting feedback from the LLM to ideate approaches that fit for the problem. We then ensemble the diverse prompts in our method DIV-SE (DIVerse reasoning path Self-Ensemble) across multiple inference calls. We also propose a cost-effective alternative where diverse prompts are used within a single inference call; we call this IDIV-SE (In-call DIVerse reasoning path Self-Ensemble). Under a fixed generation budget, DIV-SE and IDIV-SE outperform the previously discussed baselines using both GPT-3.5 and GPT-4 on several reasoning benchmarks, *without modifying the decoding process.* Additionally, DIV-SE advances state-of-the-art performance on recent planning benchmarks (Valmeekam et al., 2023). Our results shed light on how to improve the Pareto frontier of the accuracy-cost trade-off.

## 1 INTRODUCTION

Large language models (LLMs) exhibit state-of-the-art performance across a myriad of tasks, with their effectiveness strongly influenced by prompt design (Anil et al., 2023; OpenAI, 2023b). For complex reasoning tasks, designing the right prompt can enable LLMs to capitalize on task structure, such as through being 'state aware' or through decomposing the problem in a tractable way. However, existing methods to design prompts are either heuristic, relying on iterative trial-and-error (White et al., 2023), or computationally costly (Lester et al., 2021).

Previous works identified two simple, yet general principles to effectively prompt LLMs and improve their performance: (i) decomposing their reasoning into individual 'thoughts' (reasoning steps), and (ii) increasing the stochasticity during decoding. Techniques like Chain-of-Thought (CoT) prompting (Wei et al., 2022) guide the LLM to walk through its reasoning steps, which significantly improves the performance. Further, self-consistency (SC) (Wang et al., 2023) methods increase the stochasticity through modifying the decoding process and obtaining multiple completions, and the resulting diversity in the reasoning process provides improvements.

However, combining the two principles raises limitations. First, inference becomes significantly more expensive due to numerous runs, each generating long completions with many reasoning steps. Further, it may be impermissible to modify the decoding process in some settings, such as commercial deployments. Finally, stochasticity-based methods do not directly guide the diversity at the level of thought or method, but rather at the token level.

In this paper, we explore how explicitly to promote the *diversity of thought* while mitigating the aforementioned issues. Prior work by Li et al. (2023) is the first to highlight the importance of prompt diversity, but their notion of diversity is variety in the demonstrations (shots) provided as part of their prompt; ours focuses more on the reasoning technique. We first solicit the LLM to produce multiple-high-level directions (which we refer to as approaches) for problem-solving (e.g., method of

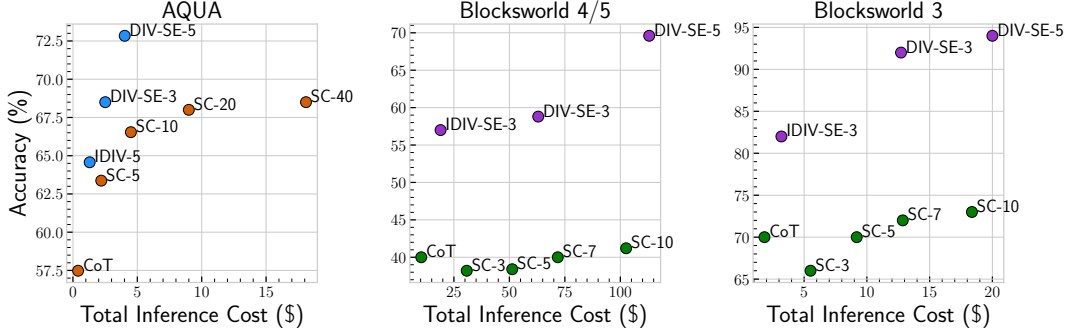

Figure 1: **Diversity of Thought enhances the inference cost and accuracy trade-off.** We compare DIV-SE and IDIV-SE with SC (Wang et al., 2023) and CoT (Wei et al., 2022) across three benchmarks. Panels show (i) AQUA-RAT on GPT-3.5 in few-shot-CoT setting, and (ii) Blocksworld 3 and 4/5 on GPT-4 in zero-shot-CoT setting. The x-axis indicates the total cost (as defined in § 3) of running inference with the LLM on the benchmark with the given method, the y-axis indicates the LLM's performance. Colors are used to distinguish between settings (few-shot-CoT and zero-shot-CoT) and methods (our proposed techniques - DIV-SE/IDIV-SE, standard CoT, and self-consistency method). Blue and orange represent the few-shot-CoT settings for DIV-SE/IDIV-SE and the self-consistency method, respectively. Similarly, purple and green are used to represent the zero-shot-CoT settings for DIV-SE/IDIV-SE and the self-consistency method, respectively.

elimination, visualization techniques etc. for math reasoning problems). We then leverage GPT-4 to style-transfer examples used in prior work (Wei et al., 2022) into the corresponding approaches[1].

Leveraging diverse approaches, we propose DIV-SE (DIVerse reasoning path Self-Ensemble) to extract and aggregate responses across multiple inference calls (§ 2.2). Since these distinct approaches introduce diversity at the thought level, our methodology results in improved ensemble accuracy. In Fig. 1, we show that it yields more accurate results across multiple reasoning benchmarks at a fixed inference cost, without modifying the decoding procedure. For instance, in the Blocksworld 4/5 task (Valmeekam et al., 2022), diversity of thought improves the performance by 29.6 percentage points. However, this method still leverages multiple inference calls, which could be costly.

To further reduce inference costs, we build on the observation that the approaches are often mutually independent, and can be combined in a single prompt to solicit multiple solutions. Based on this premise, we propose IDIV-SE (In-call DIVerse reasoning path Self-Ensemble; § 2.2), which combines $n$ approaches within the same prompt and aggregates the $n$ resulting outputs to leverage diversity with a reduced cost. Fig. 1 demonstrates that this method obtains comparable accuracy to DIV-SE and obtains better performance than prior work for lower inference costs.

Overall, across multiple domains and reasoning tasks (§ 3), we push the pareto frontier of the cost-accuracy trade-off of prompting strategies, outperforming both CoT and SC prompting on both GPT-3.5 and GPT-4. This is evident from Fig. 1 for the AQUA-RAT benchmark (Ling et al., 2017), where there is a performance improvement of 16.52 percentage points.

# 2 SOLICITING DIVERSITY THROUGH LLM INTERACTIONS

## 2.1 USING LLM AS A GUIDE TO DESIGN DIVERSE APPROACHES

LLMs trained on internet-scale data encode a significant amount of knowledge from multiple domains (Liang et al., 2022; Bubeck et al., 2023). Even though LLMs may not be perfect at solving reasoning tasks per se, we hypothesize that they may still be helpful in providing high-quality feedback. Here, we use LLMs to guide the design of potential approaches for complex reasoning.

**Step 1. Extracting Approaches & Personas.** We wish to solicit feedback from the LLM on how to solve tasks. We term this process DIVERSEPROMPTING. To do so, we utilize the following methodol-

---

[1] We do this to ensure a fair comparison between prior work and us.

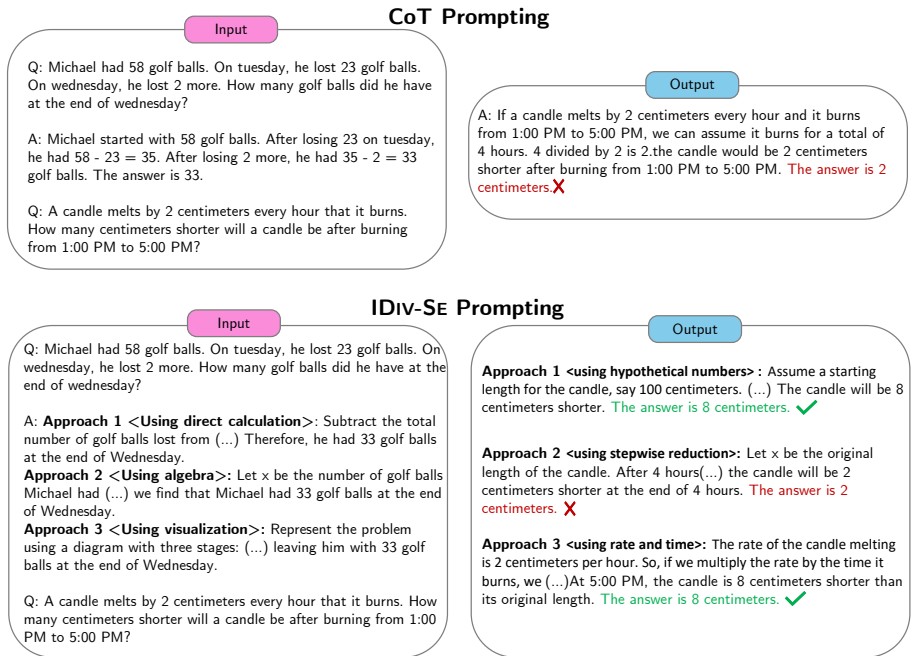

Figure 2: **Diversity of Thought**. This illustration depicts CoT and IDɪv-Sᴇ prompting strategies. Notice that both have a single example. However, IDɪv-Sᴇ presents more diversity in terms of reasoning paths. This enables it to generate diverse completions, yielding more accurate responses.

ogy: (i) We randomly pick a question $q$ from the reasoning task we want to evaluate. (ii) We create an instruction prompt $i$ where we ask the LLM to generate the names of $n \in [1, 5]$ *approaches* to solve the aforementioned question; (iii) We also provide a template $t$ that the LLM must conform to while generating the approaches. Thus, the overall prompt is $p = < i||q||t >$; (iv) We get the responses from the LLM $f$ i.e., $o = f(p)$. The final prompt used for this process is visualized in Fig. 6. We extract the part of the response that is compliant with the prescribed template and store it. We repeat this process $m$ times, to obtain a total of $m \cdot n$ candidate approaches[2]. We build a word cloud using these approaches and pick the top 5 approaches (based on frequency of occurrence). For example, for the GSM8K task, some of the LLM-generated approaches include: using visualizations, working backwards, using direct calculation, and method of elimination.

In addition to specifying an approach for "how" to solve a reasoning problem, specifying a persona (e.g., "Think like Alan Turing") can also influence how the LLM behaves. For instance, the impact of personas was similarly noted in prior work (Salewski et al., 2023). One can repeat the above process used to extract approaches to instead extract relevant personas for a given reasoning task. However, our simplified approach asked the model directly for relevant personas for a given task and then included them in the set of candidate personas $P$ used for the final set of prompts.

**Step 2. Choosing the Best Persona, Approach Pair.** The choice of persona and approaches introduces a principled way to promote diversity. Assume the set of personas is $P$, and the set of approaches is $A$. The Cartesian product of the set of personas $P$ and the power-set of approaches $2^{|A|}$ yields a lower bound on the total number of prompts one could generate. In practice, for all (persona, approach) combinations, we evaluate the prompt formed using the composition on a held-out set and choose those with the highest performance. In the scenario where the best approaches come from different personas e.g., $\{(P_1, A_1), (P_2, A_1), (P_2, A_2)\}$, we pick the (persona, approach) pairs with the persona which has the most approaches (i.e., $\{(P_2, A_1), (P_2, A_2)\}$). We perform this process once (for GPT-3.5 Turbo), and re-use our selection across all LLMs we evaluate.

**Step 3. Style Transfer to Create Demonstrations.** Once the approaches are fixed, we ask the LLM to modify existing demonstrations with the given set of approaches. Specifically, we take the

---

[2]In practice, we set $m = 100$ and $n = 5$.

demonstrations provided in prior work (Wei et al., 2022), and ask the LLM to solve them in the style of a chosen approach; we term this an *augmented demonstration*. For instance, for five approaches and a given demonstration, we will have five augmented demonstrations. This is visualized in the bottom left of Fig. 2, where the prompt contains different approaches to solve a math problem.

## 2.2 DESIGNING THE PROMPTS

We now describe two techniques to generate prompts with the demonstrations we have accumulated.

**DIV-SE:** We first propose DIV-SE (DIVerse reasoning path Self-Ensemble), a method to execute a diverse set of approaches in different inference calls and aggregate their solutions. Apart from the question to be solved and the augmented demonstrations, the final prompt contains a single persona and additional instructions. An example can be visualized in Fig. 7. Diversity is ensured through running inference with multiple prompts, each with a different augmented demonstration. However, since the approaches are executed separately, generating a solution (via aggregation of multiple responses) requires multiple inference calls, which could be costly.

**IDIV-SE:** To further reduce the inference costs while promoting diversity, we propose IDIV-SE (In-call DIVerse reasoning path Self-Ensemble). In IDIV-SE, the final prompt is a composition of "all" augmented demonstrations, the question to be solved, and contains a single persona. An example can be visualized in Fig. 2 (bottom left). This noticeably decreases the number of calls to be made, since all demonstrations are presented within the same prompt. We note that there might be error propagation due to the autoregressive nature of models i.e., errors in generations of earlier approaches may spill over to generations of subsequent approaches. We evaluate this in detail in § 3.3.1.

We find that with the IDIV-SE method, the LLM is not limited to the approaches we provide; it also develops its own new strategies. For instance, when it is tasked to solve math problems using the approaches depicted in Fig. 2, it independently formulates new approaches (using hypothetical numbers, stepwise reduction and rate and time), that weren't part of the input prompt. This suggests the rich diversity in reasoning paths IDIV-SE induces, and its ability in coercing the LLM into following these paths.

Crucially, DIVERSEPROMPTING finds approaches that are *general and reusable* across similar reasoning problems, increasing its practicality. This also reduces the cost of repeatedly evaluating them on a separate set.

**Step 4. Aggregation.** Across both prompting strategies, we aggregate the responses via a simple majority vote. However, one could assume a smarter aggregation strategy, such as utilizing the LLM to aggregate the responses itself. In § 3.3.2, we consider an aggregation strategy proposed by Yoran et al. (2023) and describe how compatible it is with our prompting approaches.

## 3 EXPERIMENTS

**Tasks & Datasets.** We consider the following reasoning benchmarks.

1. **Arithmetic Reasoning**: We use: (i) AQUA-RAT (Ling et al., 2017), a suite of algebraic word problems, and (ii) GSM8K (Cobbe et al., 2021), a benchmark of grade-school math problems (involving elementary arithmetic operations). For both datasets, we use the test split, containing 254 and 1319 questions respectively.
2. **Planning Abilities**: We use the Planning benchmark proposed in Valmeekam et al. (2022; 2023). The benchmark consists of two datasets: one involves 3 blocks and consists of 100 instances, while the other dataset involves up to 5 blocks and consists of 500 instances.
3. **Commonsense Reasoning**: We use CommonsenseQA (Talmor et al., 2019) which consists of generic multiple-choice questions elicited for testing common sense reasoning.

Note that we do not explicitly test for dataset contamination. While it is known that OpenAI (2023b) trained GPT-4 on a subset of GSM8K, little else is known about its training data. Prior work has also not presented a detailed contamination analysis. We stress that the emphasis on our work is to show *relative improvements* using our technique in comparison to others.

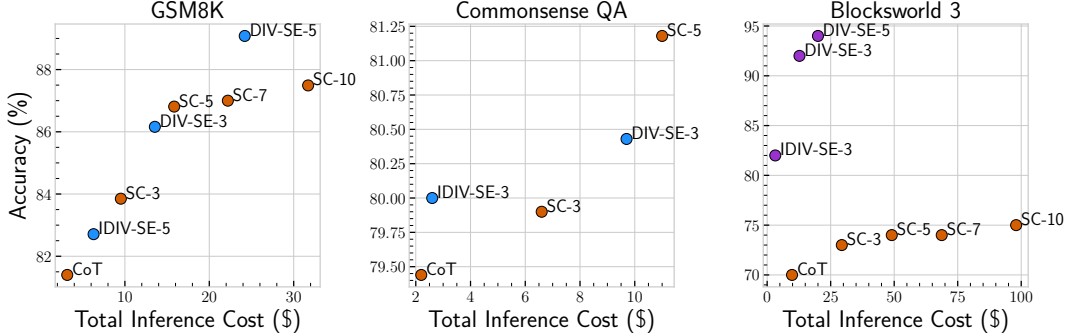

Figure 3: **Diversity of Thought enhances the inference cost and accuracy trade-off.** We compare DIV-SE and IDIV-SE with SC (Wang et al., 2023) and CoT (Wei et al., 2022) across three benchmarks. Panels show (i) GSM8K and CommonsenseQA on GPT-3.5 in few-shot-CoT setting, and (ii) Blocksworld 3 on GPT-4 (where only SC-$s$ is in few-shot-CoT setting). The x-axis indicates the total cost (as defined in § 3) of running inference with the LLM on the benchmark with the given approach, the y-axis indicates the LLM's performance. Notice that for Blocksworld 3, despite being in the zero-shot-CoT setting, our approaches are more performant than the SC-$s$ (few-shot-CoT) baseline. Colors are used to distinguish between settings (few-shot-CoT and zero-shot-CoT) and methods (our proposed techniques - DIV-SE, IDIV-SE, standard CoT, and self-consistency method). Blue is used to show the few-shot-CoT results of our proposed technique, while orange is used for the few-shot-CoT settings of the self-consistency method. Purple is used to represent the zero-shot-CoT results of our proposed technique for the Blocksworld 3 task.

**Language Models.** We evaluate our proposed methods on both GPT-3.5 Turbo (OpenAI, 2022) and GPT-4 (OpenAI, 2023a). We also conduct an additional evaluation on LLaMA-2 70B (Touvron et al., 2023) to explore the performance of our technique on open-source LLMs. For the latter, we use `meta-llama/Llama-2-70b-chat-hf` through the Transformers library (Wolf et al., 2019).

**Baselines.** We consider Chain-of-Thought (CoT) (Wei et al., 2022) and Self-Consistency (SC) (Wang et al., 2023) as our baselines. For CoT, we consider two settings: zero-shot-CoT (Kojima et al., 2022) (i.e., "Think step by step" is added to the prompt), and few-shot-CoT (i.e., CoT with demonstrations).

In our SC runs, we set the temperature $T = 0.7$ without top-$k$ truncation and sample up to $s \in [1, 10]$ outputs (denoted SC-$s$). For all other approaches, we set $T = 0$. We use ensembles of size 5 in IDIV-SE and DIV-SE for GSM8K and AQuA. For the planning and commonsense benchmarks, we use a size of 3.

**Performance Metrics.** We measure the accuracy on the task, and the inference cost associated with generation. Note that average accuracy is the average across all possible combinations of ensembles. To measure the cost, we assume 1000 tokens are about 750 words[3]. For GPT-4 (8K) the input and output prices used to estimate inference cost are \$0.03/1k tokens and \$0.06/1k tokens, respectively. For GPT 3.5 Turbo (16K), the input and output prices used in the cost estimation are \$0.003/1k (tokens) and \$0.004/1k (tokens) respectively.

Salient features of our results include:

1. For the challenging planning benchmark (Blocksworld 4/5), our techniques improve accuracy by 29.6 percentage points achieving state-of-the-art performance.
2. Across most benchmarks we consider, our techniques provide substantial performance gains. They are also Pareto optimal (in terms of the utility vs. cost trade-off). Using GPT-4 for Blocksworld 3, our approach (in the zero-shot-CoT setting) setting is substantially more effective than SC-10 (in the few-shot-CoT setting) at $4\times$ lower cost (Figure 3 (rightmost figure)).
3. Since prompts are chained together in IDIV-SE, error propagation is possible. Our evaluation on AQUA-RAT suggests that this is minimal (less than 6%).
4. When combined with aggregation approaches that are capable of reasoning across the diverse generations (Yoran et al., 2023), we observe additional performance gains. When evaluated on

---

[3]https://openai.com/pricing

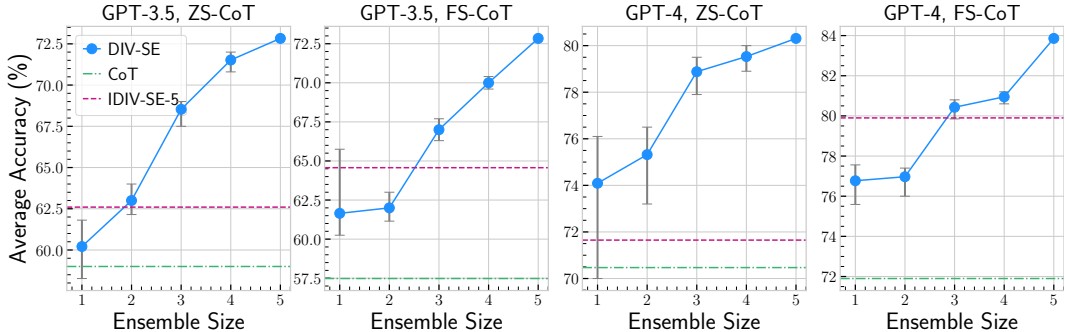

Figure 4: **Average accuracy** for different ensemble sizes on AQUA-RAT for zero-shot-CoT and few-shot-CoT settings on GPT-4 and GPT-3.5. Note that all graphs are zoomed in.

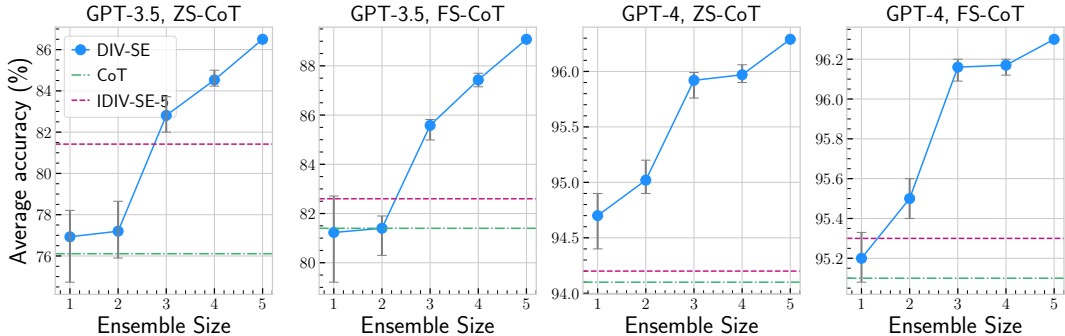

Figure 5: **Average accuracy** for different ensemble sizes on GSM8K for zero-shot-CoT and few-shot-CoT settings on GPT-4 and GPT-3.5. Note that all graphs are zoomed in.

the AQUA-RAT benchmark, we see an accuracy of 67.7% for GPT-3.5 (a 3.23 percentage point improvement to the majority voting baseline).

## 3.1 MAIN RESULTS

### 3.1.1 AQUA-RAT

**GPT-4 Results:** From Fig. 4 (right), we see that DIV-SE achieves an accuracy increase of 9.84 and 14.6 percentage points (p.p) in the few-shot-CoT (baseline accuracy of 71.9%) and zero-shot-CoT (baseline accuracy of 70.47%) settings, respectively. While the gains from IDIV-SE are nominal in the zero-shot-CoT configurations, it achieves a boost of 7.7 p.p in the few-shot-CoT setting.

**GPT-3.5 Results:** From Fig. 4 (left), we see that DIV-SE yields a gain of 14.23 and 16.52 p.p in the few-shot-CoT (baseline of 57.48%) and zero-shot-CoT (baseline accuracy of 59%) settings, respectively. Within the few-shot-CoT setting, IDIV-SE gets an absolute accuracy increase of 7 p.p.

Note that Fig. 1 also displays the total inference cost. Both IDIV-SE and DIV-SE are Pareto optimal, indicating their capacity to achieve a higher accuracy while maintaining low costs.

### 3.1.2 GSM8K

**GPT-4 Results:** As shown in Fig. 5, accuracy on GSM8K have nearly plateaued, with the zero-shot-CoT and few-shot-CoT baselines achieving accuracies of 94% and 95% respectively. IDIV-SE does not produce any significant gains in either setting. On the other hand, DIV-SE reaches accuracy of 96.3% in both few-shot-CoT and zero-shot-CoT settings, providing a modest improvement.

**GPT-3.5 Results:** Here, the gains are more substantial. Compared to the zero-shot-CoT baseline of 76.11%, IDIV-SE provides an accuracy improvement of 5.31 p.p. DIV-SE goes a step further,

enhancing the accuracy by 10.39 p.p. In the few-shot-CoT setting, DIV-SE posts an accuracy improvement of 7.68 p.p (with a baseline accuracy of 82.6%).

Fig. 5 (left) presents the cost vs. accuracy trade-offs between IDIV-SE, DIV-SE, and SC. While the performance of SC does improve with the expansion of reasoning paths, both IDIV-SE and DIV-SE offer better trade-offs.

### 3.1.3 PLANNING - BLOCKSWORLD DOMAIN

**Setup:** The benchmark provides both natural language and Planning Definition and Domain Language prompts. We use natural language prompts in all the experiments. For the baseline runs, we introduce minor alterations to the prompt originally proposed by Valmeekam et al. (2023). These changes involve incorporating an explicit directive to prevent under-block movement and resolving minor language ambiguities we observed to be problematic during initial investigation. Furthermore, we reposition the initial condition and goal state information to the beginning of the prompt. The modified improved prompt is presented in Fig. 8.

We aggregate the plans through majority voting and utilize string matching for comparing the plans. As a result, we optimize the plan by eliminating the redundant'no-op' steps.

**GPT-4 Results:** We note that GPT-4 performs slightly better in a zero-shot setting, and use this to run all experiments. From Fig. 1, notice that for the Blocksworld 3 case, zero-shot-CoT records an accuracy of 70%, while SC-10 reaches an accuracy level of 73%. IDIV-SE enhances the absolute accuracy by 12 p.p above the zero-shot-CoT baseline, while DIV-SE produces an impressive state-of-the-art accuracy of 94%. An analysis of the six unsuccessful instances suggests the capacity for further performance improvement by increasing the size of the ensemble, as already two out of five current approaches generate accurate plans. For the Blocksworld 4/5 case, the zero-shot-CoT accuracy is 40%, while SC-10 has an accuracy of 41.2%. Here, IDIV-SE results in an absolute gain of 17 p.p above the zero-shot-CoT baseline, and DIV-SE too enhances performance, leading to an accuracy of 69.6%.

As outlined in Fig. 1 (middle + right), both IDIV-SE and DIV-SE achieve Pareto optimality.

**GPT-3.5 Results:** The baseline performance on Blocksworld 3 is 6%, and on Blocksworld 4/5 is 0.6%. We do not see any additional improvement using both IDIV-SE and DIV-SE. Qualitatively, we observe that during plan generation, GPT-3.5 fails to follow the restrictions provided as part of the problem instructions too often, leading to either infeasible or incorrect plans.

### 3.1.4 COMMONSENSEQA

Table 1 presents the results of the experiments. Overall, the improvements in accuracy are relatively modest. This is likely because answering questions in CommonsenseQA does not demand as much reasoning and thought diversity as is required in some other benchmarks. In addition, the dataset also contains a number of ambiguous questions, which if read verbatim may have many plausible answers but the ground truth contains only one answer. From Fig. 3 (right), we see that our approaches are still on the Pareto frontier, but so are the SC approaches.

| | Method | Zero-shot-CoT (%) | Few-shot-CoT (%) |
|---|---|---|---|
| | CoT | 71.4 | 79.4 |
| GPT-3.5 | IDIV-SE | 74.0 | 80.0 |
| | DIV-SE | **74.5** | **80.4** |
| | CoT | 81.6 | 87.7 |
| GPT-4 | IDIV-SE | **82.5** | **89.0** |
| | DIV-SE | 81.7 | 88.0 |

Table 1: **CommonsenseQA Results.** Notice that the performance gains are not as significant. We conjecture this to be the case due to the reduced reasoning requirement for this particular benchmark.

## 3.2 OPEN SOURCE MODELS

Due to our limited computational budget, we only performed experiments with the AQUA-RAT benchmark. See Appendix B for further details. Table 2 demonstrates the results for Llama2-70B with 8-bit quantization. DIV-SE and IDIV-SE demonstrate an improvement of over 10 p.p over the baseline in the few-shot-CoT settings. However, the gain in the zero-shot-CoT setting has been negligible. We hypothesize that this is partly due to model capabilities to follow instructions. As the models get more capable, we observe diverse reasoning also benefiting zero-shot settings (c.f. Fig. 4).

| Prompting Strategy | Zero-shot-CoT (%) | Few-shot-CoT (%) |
|---|---|---|
| CoT | 31.32 | 29.1 |
| IDIV-SE | 27.00 | 39.7 |
| DIV-SE | **32.00** | **39.9** |

Table 2: **Accuracy of AQUA-RAT on LLaMA-2 70B**. Observe that while the gains are minimal in the zero-shot setting (if any), we see a 10.8 percentage point gain in the few shot setting.

## 3.3 ABLATION STUDIES

### 3.3.1 ERRORS & PROMPT UTILITY

**Error Propagation:** Since approaches are chained together in IDIV-SE, there is a possibility for error propagation (i.e., the LLM generates an incorrect response to one approach and autoregressively propagates it forward). To quantify this, we select examples where the solution is incorrect and all five approaches produce the same erroneous answer. We focus only on these cases to see if e.g. a wrong conclusion in the initial approaches leaks into the following ones. Next, we attempt the last two approaches again in a separate session: if the LLM generates the same outcomes as in the original session (i.e., IDIV-SE setup) within 3 attempts, we consider it as no error propagation. However, if it does not produce the same answer within the 3 attempts, we interpret this as a case of error propagation since the change in answer could be attributed to the initial approaches with wrong answers in the chain. We measure this phenomenon on AQUA-RAT in the few-shot-CoT setting on both GPT-4 and GPT-3.5. We find that GPT-4 and GPT-3.5 have error propagation rates of 6.2% and 5.5% respectively. Reducing these error rates remains a challenging problem given the autoregressive nature of current LLMs. Future work could envision a decrease in error rates by changing the prompt to encode dependencies between the approaches such that errors in the former are fixed in the latter.

| Dataset, Model | Persona, Approach | Accuracy (%) |
|---|---|---|
| AQUA-RAT, GPT-3.5 | ∅,Think step by step | 57.48 |
| | ∅, Using Algebra | 60.24 **(+2.76)** |
| | Thinking like Alan Turing, ∅ | 61.81 **(+4.33)** |
| | Dr. Patel: A renowned mathematician, ∅ | **65.75 (+8.27)** |
| Blocksworld 4/5, GPT-4 | ∅, State tracking prompt (Valmeekam et al., 2022) | 42.00 |
| | ∅, Finite State Machine | 55.80 **(+13.80)** |
| | Alan Turing, Action Rationale | 57.80 **(+15.80)** |
| | Alan Turing, Progressive Block Placement Approach | **58.80 (+16.80)** |

Table 3: **Prompts, derived from approaches and personas, boost performance.** Blue rows denote zero-shot-CoT prompts, while black lines denote few-shot-CoT prompts. ∅ denotes absence (of persona or approach respectively).

**Beyond Thinking Step by Step:** The diverse approaches and personas we utilize not only enhance the performance in IDIV-SE and IDIV-SE, but are also independently superior to zero-shot-CoT. Table 3 highlights this effect. This further highlights the importance of probing the model for suggestions via DIVERSEPROMPTING.

### 3.3.2 ALTERNATIVE AGGREGATION STRATEGIES

Our aggregation thus far relies on majority voting. Alternatively, we can also utilize the meta reasoning technique proposed by Yoran et al. (2023) to accumulate the results. This technique relies

| Method | GPT-4 (%) | GPT-3.5 (%) |
|---|---|---|
| Majority Voting | **79.90** | 64.47 |
| Meta Reasoning | 79.24 | **67.70** |

Table 4: **Alternative aggregation strategies.** Observe that, for the AQUA-RAT benchmark (in the few-shot-CoT setting), IDIV-SE produces more accurate results only with GPT-3.5.

on exploiting the rich information present in the reasoning steps generated. To this end, we store the responses generated by IDIV-SE, and request the model to meta reason over them in a different prompt (i.e., different session). The results in Table 4 suggest that the reasoning paths proposed contain rich information that is exploited by the meta reasoning aggregation mechanism for both models, albeit nominally for GPT-4. Future post-hoc techniques may even consider to learn about the accuracy of the diverse prompting approaches, and weigh them accordingly. Nevertheless, the fact that techniques presented here provide visible improvements even with simply majority vote, demonstrates their added value independently from aggregation algorithms.

## 4 RELATED WORK

**Prompt Optimization:** Pryzant et al. (2023) models the prompts as optimizable (albeit discrete) variables, and minimizes the loss of the reasoning task. Jones et al. (2023) optimize over the prompt space, but to identify failure modes. However, optimization-based approaches often require the task to have a differentiable loss function, which is a strong condition. In our work, we utilize feedback from the LLM (not through gradients) in helping design the prompt. Cheng et al. (2023) define an approach to batch the responses for multiple queries within a prompt; IDIV-SE is inspired by this.

**Decoding Optimizations and Tools:** Wang et al. (2023) replace the naive greedy decoding by sampling a diverse set of reasoning paths (e.g., through temperature sampling), and then selects the most consistent answer. Chen et al. (2022) express the reasoning process as a program, which is then delegated to an external tool. Retrieval augmented generation (e.g., Shuster et al. (2021)) also relies on a similar premise (i.e., the existence of a trusted tool – the retriever – to facilitate accurate generation). In our work, we neither change the decoding process nor assume the existence of trusted tools. This makes our solution directly applicable to black-box models.

**Prompting Strategies:** Brown et al. (2020) note that demonstrations to prompts, encoded as input-output pairs, produce drastic performance increase in larger LLMs. Wei et al. (2022) encourage internal dialogue by forcing the LLM to generate a sequence of intermediate steps for reasoning problems. This improves reasoning performance on larger LLMs (Nye et al., 2021; Chung et al., 2022; Kojima et al., 2022). Zhou et al. (2022) go a step further; they (automatically) break a complex problem into simpler sub-problems and then solve them in sequence. Across all these techniques, the common practice is to keep the prompts fixed, but aggregate responses across multiple trials of them by varying the temperature. In our work, we vary the input prompt itself. A work that is similar in spirit is that of Yoran et al. (2023), which instead of aggregating the response of multiple reasoning paths, forces the model to reason across them before aggregation. Another relevant work is that of Li et al. (2023), which shows the importance of prompt diversity. However, they rely on selecting few-shot demonstrations from a hold-out set (which defines diversity in their method), without explicitly stating reasoning pathways. Our work does the latter.

## 5 CONCLUSIONS

In this work, we explored promoting diversity as a principled prompting strategy. We proposed methodologies that leverage the model as a guide to design a diverse set of approaches and solutions. We further demonstrated how promoting diversity can improve the Pareto frontier of accuracy-cost trade-off and yield state-of-the-art solutions for planning tasks. Given our results, we believe that there is a large room for improvement in using the LLM as a guide to improving the prompt. While we try simple aggregation techniques, improved ensembling techniques could further utilize the diversity of the solutions, e.g. different solutions could be more reliable in different problems. Overall, our results make a case for diversity as a strong principle for designing effective prompts.

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
