# APPENDIX

## A    PROMPT USED FOR DIVERSEPROMPTING

Use five distinct approaches to solve the given problem accurately. If there is no exact match choose the closest option.
Q: {Question}
Use the following output format:
Approach 1 < name of the approach > : < Details of Approach 1 >
Approach 2 < name of the approach > : < Details of Approach 2 >
Approach 3 < name of the approach > : < Details of Approach 3 >
Approach 4 < name of the approach > : < Details of Approach 4 >
Approach 5 < name of the approach > : < Details of Approach 5 >

Figure 6: **Prompt template** for extracting diverse approaches for problem solving.

## B    MODEL DETAILS

### B.1    OPEN-SOURCE MODELS

We perform the Llama-2 70B experiments with a single 80GB A100 GPU. To fit the 70B model to a single A100, we use 8-bit precision through `bitsandbytes` (Dettmers et al., 2022a;b). Further, Dettmers et al. (2022a) reports no performance drop with this quantization method.

As the system prompt, we use `You are a helpful, respectful and honest assistant.` We perform inference with greedy decoding, having temperature $\tau = 0$.

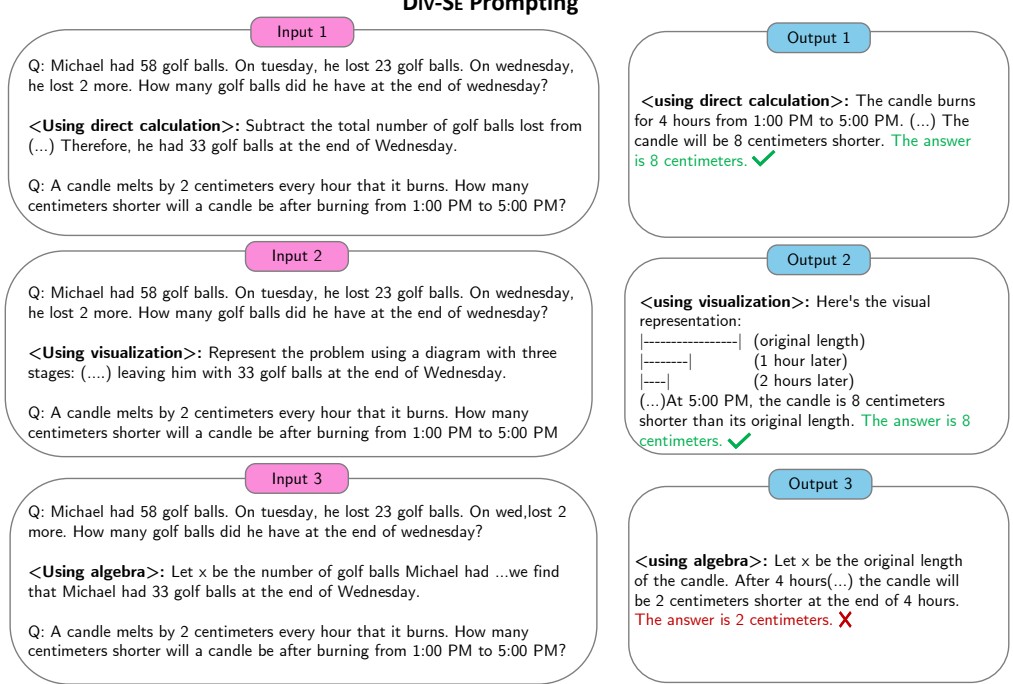

Figure 7: DIV-SE prompting

# C  RESULTS

| | Method | Zero-shot-CoT (%) | Few-shot-CoT (%) |
|---|---|---|---|
| | CoT | 76.11 | 81.40 |
| GPT-3.5 | IDIV-SE-5 | 81.42 | 82.60 |
| | DIV-SE-1 | 76.93 | 81.23 |
| | DIV-SE-2 | 77.20 | 81.40 |
| | DIV-SE-3 | 82.81 | 85.58 |
| | DIV-SE-4 | 84.53 | 87.43 |
| | DIV-SE-5 | **86.50** | **89.08** |
| | CoT | 94.10 | 87.70 |
| GPT-4 | IDIV-SE-5 | 94.20 | 95.30 |
| | DIV-SE-1 | 94.70 | 95.20 |
| | DIV-SE-2 | 95.02 | 95.50 |
| | DIV-SE-3 | 95.92 | 96.16 |
| | DIV-SE-4 | 95.97 | 96.17 |
| | DIV-SE-5 | **96.29** | **96.30** |

Table 5: Average accuracy for different ensemble sizes on GSM8K for zero-shot-CoT and few-shot-CoT settings on GPT-4 and GPT-3.5.

| | Method | Zero-shot-CoT (%) | Few-shot-CoT (%) |
|---|---|---|---|
| | CoT | 59.00 | 57.48 |
| GPT-3.5 | IDIV-SE-5 | 62.60 | 64.57 |
| | DIV-SE-1 | 60.21 | 61.65 |
| | DIV-SE-2 | 63.00 | 62.00 |
| | DIV-SE-3 | 68.53 | 67.00 |
| | DIV-SE-4 | 71.52 | 70.00 |
| | DIV-SE-5 | **72.83** | **72.84** |
| | CoT | 70.47 | 79.90 |
| GPT-4 | IDIV-SE-5 | 71.65 | 76.77 |
| | DIV-SE-1 | 74.09 | 76.77 |
| | DIV-SE-2 | 75.32 | 76.97 |
| | DIV-SE-3 | 78.88 | 80.43 |
| | DIV-SE-4 | 79.53 | 80.95 |
| | DIV-SE-5 | **80.31** | **83.86** |

Table 6: Average accuracy for different ensemble sizes on AQUA-RAT for zero-shot-CoT and few-shot-CoT settings on GPT-4 and GPT-3.5.

# D  PROMPT TEMPLATES

I am playing with a set of blocks where I need to arrange the blocks into stacks.

[STATEMENT]
As initial conditions I have that, the orange block is clear, the hand is empty, the blue block is on top of the red block, the orange block is on top of the blue block and the red block is on the table. My goal is to have that the red block on top of the blue block and the orange block on top of the red block.

Here are the actions I can do:
Pick up a block from the table
Unstack a block from on top of another block
Put down a block on the table
Stack a block on top of another block

I have the following restrictions on my actions:
I can only pick up or unstack one block at a time.
I can only pick up or unstack a block if my hand is empty.
I can only pick up a block if the block is on the table and the block is clear. A block is clear if the block has no other blocks on top of it and if the block is not picked up.
I can only unstack a block from on top of another block if the block I am unstacking was really on top of the other block.
I can only unstack a block from on top of another block if the block I am unstacking is clear.
Once I pick up or unstack a block, I am holding the block.
I can only put down a block that I am holding.
I can only stack a block on top and not under of another block if I am holding the block being stacked.
I can only stack a block on top and not under of another block if the block onto which I am stacking the block is clear.
Once I put down or stack a block, my hand becomes empty.
Once you stack a block on top of a second block, the second block is no longer clear.

What is the plan to achieve my goal? Just give the actions in the plan.

[PLAN]

Figure 8: Zero-shot prompt used in the baseline run of the Planning - Blocksworld Domain

You are playing with a set of blocks where you need to arrange the blocks into stacks. What is the plan to achieve the goal?

<Initial State> : As initial conditions you have that, the orange block is clear, the hand is empty, the blue block is on top of the red block, the orange block is on top of the blue block and the red block is on the table.
<Goal State> : Your goal is to have that the red block on top of the blue block and the orange block on top of the red block.

Here are the actions you can do:
-Pick up a block from the table
-Unstack a block from on top of another block
-Put down a block on the table
-Stack a block on top of another block

Rules:
1. You can only pick up or unstack one block at a time.
2. You can only pick up or unstack a block if your hand is empty.
3. You can only pick up a block if the block is on the table and the block is clear. A block is clear if the block has no other blocks on top of it and if the block is not picked up.
4. You can only unstack a block from on top of another block if the block you are unstacking was really on top of the other block.
5. You can only unstack a block from on top of another block if the block you are unstacking is clear.
6. Once you pick up or unstack a block, you are holding the block.
7. You can only put down a block that you are holding.
8. You can only stack a block on top and not under of another block if you are holding the block being stacked.
9. You can only stack a block on top and not under of another block if the block onto which you are stacking the block is clear.
10. Once you put down or stack a block, your hand becomes empty.
11. Once you stack a block on top of a second block, the second block is no longer clear.

Using a finite state machine and a search algorithm what is the plan to achieve the goal? You can model each state of the blocks configuration on the table and the hand as a state. For each action step check that the step follows the rules and that the step brings you closer to the goal. After each action describe the state of the table and hand. Always check whether the final state satisfies the goal mentioned. <Goal State> : Your goal is to have that the red block on top of the blue block and the orange block on top of the red block.

[PLAN]

Figure 9: The Zero-shot prompt using Finite State Machine Approach for solving the Planning - Blocksworld Domain Problem.

You are playing with a set of blocks where you need to arrange the blocks into stacks.

<Initial State> : As initial conditions you have that, the orange block is clear, the hand is empty, the blue block is on top of the red block, the orange block is on top of the blue block and the red block is on the table.

<Goal State> : Your goal is to have that the red block on top of the blue block and the orange block on top of the red block.

Here are the actions you can do:
-Pick up a block from the table
-Unstack a block from on top of another block
-Put down a block on the table
-Stack a block on top of another block

Rules:
1. You can only pick up or unstack one block at a time.
2. You can only pick up or unstack a block if your hand is empty.
3. You can only pick up a block if the block is on the table and the block is clear. A block is clear if the block has no other blocks on top of it and if the block is not picked up.
4. You can only unstack a block from on top of another block if the block you are unstacking was really on top of the other block.
5. You can only unstack a block from on top of another block if the block you are unstacking is clear.
6. Once you pick up or unstack a block, you are holding the block.
7. You can only put down a block that you are holding.
8. You can only stack a block on top and not under of another block if you are holding the block being stacked.
9. You can only stack a block on top and not under of another block if the block onto which you are stacking the block is clear.
10. Once you put down or stack a block, your hand becomes empty.
11. Once you stack a block on top of a second block, the second block is no longer clear.

Thinking like Alan Turing starting from the <Initial State> build a plan to get to the <Goal State>. For each action step carefully check that the step follows the rules. <Goal State> : Your goal is to have that the red block on top of the blue block and the orange block on top of the red block.

output format for each step until you reach the goal state:
<state> : <state>
<action> : < action to be performed in this step >
<assess the action> : < are we building the stack bottom up, check carefully>

Figure 10: The Zero-shot prompt used with the persona of Alan Turing and Action Rationale approach for solving the Planning - Blocksworld Domain Problem.

You are playing with a set of blocks where you need to arrange the blocks into stacks.

<Initial State> : As initial conditions you have that, the orange block is clear, the hand is empty, the blue block is on top of the red block, the orange block is on top of the blue block and the red block is on the table.
<Goal State> : Your goal is to have that the red block on top of the blue block and the orange block on top of the red block.

Here are the actions you can do:
-Pick up a block from the table
-Unstack a block from on top of another block
-Put down a block on the table
-Stack a block on top of another block

Rules:
1. You can only pick up or unstack one block at a time.
2. You can only pick up or unstack a block if your hand is empty.
3. You can only pick up a block if the block is on the table and the block is clear. A block is clear if the block has no other blocks on top of it and if the block is not picked up.
4. You can only unstack a block from on top of another block if the block you are unstacking was really on top of the other block.
5. You can only unstack a block from on top of another block if the block you are unstacking is clear.
6. Once you pick up or unstack a block, you are holding the block.
7. You can only put down a block that you are holding.
8. You can only stack a block on top and not under of another block if you are holding the block being stacked.
9. You can only stack a block on top and not under of another block if the block onto which you are stacking the block is clear.
10. Once you put down or stack a block, your hand becomes empty.
11. Once you stack a block on top of a second block, the second block is no longer clear.

Thinking like Alan Turing, starting from the <Initial State> build a plan to get to the <Goal State> . For each action step carefully check that the step follows the rules. Divide the task into smaller steps, starting with placing the bottom block first, followed by the middle blocks, and finally the top block. <Goal State> : Your goal is to have that the red block on top of the blue block and the orange block on top of the red block.

[PLAN]

Figure 11: The Zero-shot prompt used in the Alan Turing + Progressive Block Placement Approach for solving the Planning - Blocksworld Domain Problem.

As a math professor, you will solve the given problem accurately '<using algebra>'. If there is no exact match choose the closest option.

Question: John found that the average of 15 numbers is 40. If 10 is added to each number then the mean of the numbers
Answer Choices: (A) 50 (B) 45 (C) 65 (D) 78 (E) 64
Approach: <using algebra>
Let's denote the sum of the 15 numbers as S. Since the average of the 15 numbers is 40, we can write the equation: S / 15 = 40 S = 40 * 15 = 600 Now, we add 10 to each number, so the new sum is S + 15 * 10 = 600 + 150 = 750. The new average is: New average = 750 / 15 = 50
The answer is: (A) 50

Question: If a / b = 3/4 and 8a + 5b = 22,then find the value of a.
Answer Choices: (A) 1/2 (B) 3/2 (C) 5/2 (D) 4/2 (E) 7/2
Approach: <using algebra>
To solve the given problem, we can use substitution. Since a / b = 3/4, we can write that as: a = 3b/4 Now, substitute this expression for a into the second equation: 8(3b/4) + 5b = 22 Simplify and solve for b: 6b + 5b = 22 11b = 22 b = 2 Now that we have the value of b, we can find the value of a: a = 3b/4 a = 3(2)/4 a = 6/4 a = 3/2 So, the value of a is (B) 3/2.
The answer is: (B) 3/2

Question: A person is traveling at 20 km/hr and reached his destiny in 2.5 hr then find the distance?
Answer Choices: (a) 53 km (b) 55 km (c) 52 km (d) 60 km (e) 50 km
Approach: <using algebra>
Using the formula distance = speed × time, we can calculate the distance as follows: Distance = 20 km/hr × 2.5 hr = 50 km So, the closest answer choice is (e) 50 km.
The answer is: (e) 50 km

Question: How many keystrokes are needed to type the numbers from 1 to 500?
Answer Choices: (a) 1156 (b) 1392 (c) 1480 (d) 1562 (e) 1788
Approach: <using algebra>
Let's break down the number of keystrokes needed into groups based on the number of digits: One-digit numbers (1-9): There are 9 one-digit numbers, so we need 9 keystrokes. Two-digit numbers (10-99): There are 90 two-digit numbers, each requiring 2 keystrokes, so we need 90 * 2 = 180 keystrokes. Three-digit numbers (100-500): There are 401 three-digit numbers (500 - 100 + 1), each requiring 3 keystrokes, so we need 401 * 3 = 1203 keystrokes. Now let's add up the keystrokes from all groups: 9 + 180 + 1203 = 1392. Therefore, the closest option is (b) 1392.
The answer is: (b) 1392

Figure 12: AQuA few-shot-CoT prompt using algebraic approach.

Think like Alan Turing and solve the problem.If there is no exact match choose the closest option.

Question: John found that the average of 15 numbers is 40. If 10 is added to each number then the mean of the numbers
Answer Choices: (A) 50 (B) 45 (C) 65 (D) 78 (E) 64
Approach : <think like Alan Turing>
Let's analyze the problem step by step: The average of 15 numbers is 40. So, the sum of these 15 numbers is 15 * 40 = 600. 10 is added to each number. Now, since there are 15 numbers, the total increase in the sum will be 15 * 10 = 150. The new sum of these 15 numbers will be 600 + 150 = 750. To find the new mean, divide the new sum by the total number of values, which is 15. New mean = 750 / 15 = 50. Therefore, the closest option to the new mean is (A) 50.
The answer is: (A) 50

Question: If a / b = 3/4 and 8a + 5b = 22,then find the value of a.
Answer Choices: (A) 1/2 (B) 3/2 (C) 5/2 (D) 4/2 (E) 7/2
Approach : <think like Alan Turing>
Let us analyze the given information and solve the problem step by step. We are given that a / b = 3/4, which can be written as a = (3/4) * b. We are also given the equation 8a + 5b = 22. Substitute the value of a from step 1 into the equation from step 2: 8 * (3/4) * b + 5b = 22 Simplify the equation: 6b + 5b = 22 Combine the terms and solve for b: 11b = 22 b = 2 Now that we have the value of b, let's find the value of a using the equation from step 1: a = (3/4) * b a = (3/4) * 2 Simplify to find the value of a: a = 3/2 Thus, the value of a is 3/2, which corresponds to answer choice (B).
The answer is: (B)

Question: A person is traveling at 20 km/hr and reached his destiny in 2.5 hr then find the distance?
Answer Choices: (a) 53 km (b) 55 km (c) 52 km (d) 60 km (e) 50 km
Approach : <think like Alan Turing>
To find the distance, we can use the formula: distance = speed × time. In this case, the speed is 20 km/hr and the time is 2.5 hours. Distance = 20 km/hr × 2.5 hr = 50 km So, the closest option is (e) 50 km.
The answer is: (e) 50 km

Question: How many keystrokes are needed to type the numbers from 1 to 500?
Answer Choices: (a) 1156 (b) 1392 (c) 1480 (d) 1562 (e) 1788
Approach : <think like Alan Turing>
To determine the number of keystrokes needed to type the numbers from 1 to 500, let's break it down by digit count: One-digit numbers (1-9): There are 9 one-digit numbers, each requiring 1 keystroke. Total: 9 keystrokes. Two-digit numbers (10-99): There are 90 two-digit numbers, each requiring 2 keystrokes. Total: 90 * 2 = 180 keystrokes. Three-digit numbers (100-500): There are 401 three-digit numbers, each requiring 3 keystrokes. Total: 401 * 3 = 1203 keystrokes. Adding the keystrokes for each group, we get: 9 + 180 + 1203 = 1392 keystrokes. Thus, the closest option is (b) 1392
The answer is: (b) 1392

Figure 13: AQuA few-shot-CoT prompt with the persona of Alan Turing.

Think like Dr. Patel, the mathematician who specializes in algebra and solve the problem. If there is no exact match choose the closest option.

Question: John found that the average of 15 numbers is 40. If 10 is added to each number then the mean of the numbers
Answer Choices: (A) 50 (B) 45 (C) 65 (D) 78 (E) 64
Approach : <think like Dr. Patel>
Let's consider the problem step by step. John found that the average of 15 numbers is 40. To find the sum of these 15 numbers, we can use the formula: Sum = Average × Number of elements Sum = 40 × 15 Sum = 600 Now, 10 is added to each of the 15 numbers. So, we can find the new sum by adding 10 × 15 = 150 to the original sum: New Sum = 600 + 150 New Sum = 750 To find the new average (mean), we can use the formula: New Average = New Sum ÷ Number of elements New Average = 750 ÷ 15 New Average = 50
The answer is: (A) 50

Question: If a / b = 3/4 and 8a + 5b = 22,then find the value of a.
Answer Choices: (A) 1/2 (B) 3/2 (C) 5/2 (D) 4/2 (E) 7/2
Approach : <think like Dr. Patel>
To solve this problem, first we can rewrite the equation a / b = 3/4 as a = (3/4)b. Now we can substitute this expression for a into the second equation: 8a + 5b = 22. Substitute a with (3/4)b: 8(3/4)b + 5b = 22 Simplify the equation: 6b + 5b = 22 Combine the terms: 11b = 22 Now, divide both sides by 11: b = 2 Now that we have the value of b, we can find the value of a using the equation a = (3/4)b: a = (3/4)(2) a = 3/2 The value of a is 3/2. So, the correct answer is (B) 3/2.
The answer is: (B) 3/2

Question: A person is traveling at 20 km/hr and reached his destiny in 2.5 hr then find the distance?
Answer Choices: (a) 53 km (b) 55 km (c) 52 km (d) 60 km (e) 50 km
Approach : <think like Dr. Patel>
To find the distance, we can use the formula: Distance = Speed × Time Here, the speed is given as 20 km/hr and the time taken is 2.5 hours. So, we can plug these values into the formula: Distance = 20 km/hr × 2.5 hr Distance = 50 km
The answer is: (e) 50 km

Question: How many keystrokes are needed to type the numbers from 1 to 500?
Answer Choices: (a) 1156 (b) 1392 (c) 1480 (d) 1562 (e) 1788
Approach : <think like Dr. Patel>
To determine the number of keystrokes needed to type the numbers from 1 to 500, let's break down the problem by analyzing the number of digits in each range: For numbers 1-9, there are 9 numbers with 1 digit each, requiring 9 keystrokes. For numbers 10-99, there are 90 numbers with 2 digits each, requiring 90 * 2 = 180 keystrokes. For numbers 100-499, there are 400 numbers with 3 digits each, requiring 400 * 3 = 1200 keystrokes. For the number 500, it has 3 digits, requiring 3 keystrokes. Now, we can sum up the keystrokes for each range: 9 (for 1-9) + 180 (for 10-99) + 1200 (for 100-499) + 3 (for 500) = 1392 keystrokes.
The answer is: (b) 1392

Figure 14: AQuA few-shot-CoT prompt with the persona of Dr. Patel: A renowned mathematician