# OpenReview forum: "DIVERSITY OF THOUGHT IMPROVES REASONING ABILITIES OF LARGE LANGUAGE MODELS"
_ICLR.cc/2024/Conference — Submitted to ICLR 2024_

### Official Review · Reviewer_YP6t · 2023-11-02

**Soundness:** 2 fair
**Presentation:** 3 good
**Contribution:** 2 fair
**Rating:** 5
**Confidence:** 4

**Summary:**

- The authors propose a prompting method to increase the diversity of thoughts. The authors propose a two step solution to generate diverse prompts. First, ask an LLM to generate personas and approaches to solve a task. Next, (in the DIV-SE case), provide few-shot examples using the different approaches in the prompt, and have the LLM generate answers, over which we take a majority vote.
- The authors show a number of cost/accuracy trade off plots for AQUA-RAT, Blocksworld, GSM8K and Commonsense Reasoning. The authors also show results of using various ensemble sizes for DIV-SE.
- The authors examine error propagation in the case of IDIV-SE, and other aggregation strategies beyond a simple majority vote.

**Strengths:**

- Paper is well written and the method is clear.

**Weaknesses:**

- The novelty of the work is very limited. There exists work in which diverse prompts are used to generate diverse reasoning paths: [This paper ](https://arxiv.org/pdf/2206.02336.pdf) should be cited due to its similarity.
- It is not completely clear why DIV-SE works well on Blockworlds and AQUA-RAT, but less so on GSM8K and Commonsense QA.
- IDIV is likely to have significant context length constraints.
- It’s not clear whether adding a persona increases the diversity of output. I suspect the persona does not add much to the method. This requires an ablation.
- The labels on each of the Figures could be clearer. The color codes are missing.


In conclusion, while the proposed method by the authors is clear, it appears the novelty of this work is severely limited (see above). Furthermore, IDIV seems to have a significant context length constraint, and the experiments do not show very convincing improvement across the board. Therefore I cannot recomment acceptance of this paper to ICLR.

**Questions:**

- Why are the results for Blockworld 3 in Figure 1 different from the one in Figure 3? Why is the inference cost much lower for DIV-SE-3 and DIV-SE-5 in Figure 3?
- How many trials are the experiments over?
- Can you explain at a high-level why your method is so much stronger on Blocksworld 3 and Blocksworld 4 / 5 compared to the other tasks? Can you provide some outputs of DIV-SE vs. baseline to illustrate?
- Could you clarify what the various colors mean in Figure 1 and Figure 3 (purple, green, blue, orange etc.)?

---

> ### Author Response · Authors · 2023-11-18
> **Thank you for your feedback!**
>
> [part 1]
> We thank the reviewer for their comments.  We now respond to any concerns the reviewer may have. Should the reviewer have any clarifications, we are happy to further engage. If the responses are satisfactory, we kindly request the reviewer to increase their score towards acceptance.
>
> **The novelty of the work is very limited. There exists work in which diverse prompts are used to generate diverse reasoning paths: This paper should be cited due to its similarity.**
>
> Response: We thank the reviewer for sharing this paper, but would like to strongly stress that the methodology of this paper is very different from that of ours. In the paper by Li et al., the reasoning problem’s prompt is formed by providing multiple, yet different examples depending on the test sample (refer to figure 10 in their paper), and using a step-wise verifier to correct errors made. **This requires them to have a base of examples from which they can sample few-shot examples for each prompt, a critical limitation as noted in Appendix B.1 of their paper. We do not.**
>
> The “diversity” in their prompts are reasoning paths based on self-consistency for examples from the base of examples (i.e., a hold out set); these reasoning paths are not truly diverse as they all use the same approach (as highlighted in response to reviewer PAGk) and only serve as “more examples” for the model or at best as linguistic diversity triggers (but not reasoning diversity). In contrast, our work (i) involves truly diverse (refer to response to reviewer PAGk) and “related” approaches to solve the reasoning problem (refer to Figure 2 in our work), whilst (ii) limiting the number of examples in the prompt (we use 5, Li et al. need close to 20) and (iii) do not have a step-wise verifier (which further increases the cost).
>
> **It is not completely clear [how/why] DIV-SE works well on Blockworlds and AQUA-RAT, but less so on GSM8K and Commonsense QA.**
>
> Response:  We would like to stress that the minor improvements are in scenarios where the models are already very performant in the tasks; regardless - the performance using our method is consistently high (with an average improvement of 8 percentage points) and often state-of-the-art (as in the Blocksworld case, with a 29.69 percentage point improvement). All this, with a fraction of the cost of contemporary approaches. We request the reviewer to look at the table below (in the summary link) to learn more about performance gains, and Figure 1 + 3 in the paper to learn more about the cost vs. performance trade-offs.
>
> The reasons for little improvements for some models and datasets is because of task saturation (particularly in the case of GSM8K and GPT-4). This can be for a variety of reasons, such as these tasks not needing substantial reasoning capabilities as the models become bigger, or dataset contamination. However, for tasks such as Blocksworld, reasoning is more important and this is where our method shines compared to prior approaches.
>
> Summary of the results can be found here:: https://ibb.co/68PF2Dr
>
> **IDIV is likely to have significant context length constraints.**
>
> Response: The prompts proposed in the work by Li et al. are significantly longer (with an average of 6050 tokens per prompt, based on their training data provided in https://github.com/microsoft/CodeT/tree/main/DIVERSE) compared to IDIV-SE (which has an average of 523 tokens per prompt) for the GSM8K task – almost a 12x gap. Additionally, the context length of models is increasing (as shown in most recent model releases), and we believe this should not be an issue. Additionally, we have taken measures to mitigate this potential issue. Specifically, we limit our prompts to a maximum of 3-shots, as opposed to (a) 5-7 shots typically used in standard few shot prompts (Wang et al., 2023), or (b) 20 shots as used by the work of Li et al. (https://arxiv.org/pdf/2206.02336.pdf). Please note the inference cost reported in Figure 1 and 3 accounts for both input and output token costs.
>
> **It’s not clear whether adding a persona increases the diversity of output. I suspect the persona does not add much to the method. This requires an ablation.**
>
> Response: As part of our preliminary analysis, we observed that the persona was very important in generating enhanced performance. The work of Salewski et al. that we cite highlights the importance of personas in in-context learning situations.

---

> > ### Author Response · Authors · 2023-11-18
> > **part 2**
> >
> > [part 2]
> > **Why are the results for Blockworld 3 in Figure 1 different from the one in Figure 3? Why is the inference cost much lower for DIV-SE-3 and DIV-SE-5 in Figure 3?**
> >
> > Response: The difference in the results for Blocksworld 3 in Figures 1 and 3 can be attributed to the different settings used for the self-consistency method. In both figures, IDIV-SE and DIV-SE were run in the zero-shot-CoT setting. However, the self-consistency method was run using the zero-shot-CoT setting in Figure 1, while in Figure 3, it was run in the few-shot-CoT setting. As we highlighted in Figure 3, even when operating in the zero-shot-CoT setting, our methods outperform the self-consistency method’s few-shot-CoT baseline. Thanks for the question and we will clarify this difference further in text and figure captions!
> >
> > **How many trials are the experiments over?**
> >
> > Response: All experiments were run only once, as obtaining credits required to run GPT-4 multiple times was challenging. However, we did maximize for determinism and set the temperature to 0.
> >
> > **Could you clarify what the various colors mean in Figure 1 and Figure 3 (purple, green, blue, orange etc.)?**
> >
> > Response: Colours in Figures 1 and 3 are used to distinguish between settings (few-shot-CoT and zero-shot-CoT) and methods (our proposed techniques, DIV-SE/IDIV-SE, and the self-consistency method). We will clarify this better in the draft.
> > In Figure 1, blue and orange represent the few-shot-CoT settings for DIV-SE/IDIV-SE and the self-consistency method, respectively. Similarly, purple and green are used to represent the zero-shot-CoT settings for DIV-SE/IDIV-SE and the self-consistency method, respectively.
> >
> > In Figure 3, we use blue to show the few-shot-CoT results of our proposed technique, while orange is used for the few-shot-CoT settings of the self-consistency method. Purple is used to represent the zero-shot-CoT results of our proposed technique for the Blocksworld 3 task.

---

> > > ### Author Response · Authors · 2023-11-20
> > > **Any other concerns?**
> > >
> > > Dear Reviewer,
> > >
> > > Thanks again for your review! As we near the end of the discussion period, we wanted to ask if we addressed questions you raised, and hope you find our responses useful. We would love to engage with you further if there are any remaining points.
> > >
> > > We understand that the discussion period is short, and we sincerely appreciate your time and help!

---

> > > > ### Comment · Reviewer_YP6t · 2023-11-21
> > > >
> > > > Thank you for answering my questions. Thank you also for addressing the concerns highlighted.
> > > >
> > > > I am happy with the answers with respect to performance that you provided. I have increased the score accordingly.
> > > >
> > > > With respect to Li et. al. I agree there are differences between your method and theirs, but I still believe the amount of novelty in your method is limited. Perhaps more important than novelty, I agree with reviewer "Vhwc" that the way in which this paper is written currently overstates its contribution as it does not sufficiently address past related works and highlights the similarities and differences. I believe a material change to Section 1 (Introduction) and Section 4 (Related Works) might be necessary.

---

> > > > > ### Author Response · Authors · 2023-11-22
> > > > > **Thanks for your response!**
> > > > >
> > > > > Thank you very much for your feedback. We apologize for not being able to respond to your concerns satisfactorily.
> > > > >
> > > > > Our edited paper (which is uploaded) hopefully adjusts the claims in a satisfactory manner. We're curious as to why the reviewer is apprehensive when:
> > > > >
> > > > > 1. Our work is the first to highlight the importance of thought diversity (different from prompt diversity as in Li et al.) in solving reasoning problems. Prior reported works aim to batching multiple queries together (unrelated to reasoning), or utilizing more demonstrations with per-step verification (at substantially higher query cost).
> > > > > 2. The performance numbers we report are often state-of-the-art (SOTA). This is particularly true in cases where datasets were released with explicit claims of being challenging for LLMs.
> > > > > 3. The cost for achieving these SOTA numbers is often much lesser than previously published works. This may result in rapid practical adoption.
> > > > > 4. Our evaluation is comparable to those done in previous prompting papers, which were published. We consistently outperform prior work.
> > > > > 5. We've clarified the difference against papers the reviewers have shared, and will gladly incorporate them into discussion of prior work.

---

### Official Review · Reviewer_Vhwc · 2023-11-02

**Soundness:** 3 good
**Presentation:** 2 fair
**Contribution:** 3 good
**Rating:** 3
**Confidence:** 4

**Summary:**

This paper proposes a novel prompting framework, DIV-SE that ensembles diverse prompts instead of ensembling multiple decoding outputs in self-consistency. The diverse prompts are generated by two steps: 1) instruct an LLM to generate names of approaches or names of personas. 2) generate the rationale for each exemplar and each approach by analogizing to existing CoT examples. It also proposes a cost-effective variant, IDIV-SE, to reduce the cost of multiple inference calls. Compared to self-consistency, DIV-SE and IDIV-SE set up a new Pareto frontier in performance-cost trade-off.

**Strengths:**

This paper proposes a novel variant of self-ensembling for LLMs, which doesn’t require much prompt engineering and is applicable to many tasks.
The authors experimented with the proposed method on 3 tasks and 4 datasets. The proposed method outperforms self-consistency in performance-cost trade-off on the all datasets except for CommonsenseQA.

**Weaknesses:**

The methodology part of this paper is not well written. Figure 2 is readable, but Section 2 lacks a clear high-level structure. What are the goals for extracting approaches & personas, augmented demonstrations in the proposed method? Can you describe more details about how you generated the few-shot exemplars for each task? Is it automatically generated by LLM or manually written? If they are automatically generated, what if there are errors in the generated exemplars? Do you need to verify the generated exemplars on a validation set? If you need a validation set for developing this method, it would be not fair to compare it with self-consistency in terms of cost, since self-consistency works out-of-the-box without a validation set.
Merging multiple inference calls into one was first invented and discussed in Batch Prompting[1]. Please cite it and tone down your contribution on this technique.
The paper seems to be written in a rush. Page 5 is badly formatted.


[1] Cheng, et al. Batch Prompting: Efficient Inference with Large Language Model APIs. EMNLP 2023.

**Questions:**

Maybe you can put Figure 2 on page 3 instead of page 4?
Section 2.2. “It independently formulates new approaches” -> Is it a hallucination or a feature? It looks like a hallucination to me. If this is important for achieving good performance, can you provide an ablation study based on whether to allow new approaches or not?

---

> ### Author Response · Authors · 2023-11-18
> **Thank you for your feedback**
>
> We thank the reviewer for their comments, and for finding our approach novel, and one that doesn’t require extensive prompt engineering while being applicable to many tasks.  We now respond to any concerns the reviewer may have. Should the reviewer have any clarifications, we are happy to further engage. If the responses are satisfactory, we kindly request the reviewer to increase their score towards acceptance.
>
> **The methodology part of this paper is not well written. Figure 2 is readable, but Section 2 lacks a clear high-level structure. What are the goals for extracting approaches & personas, augmented demonstrations in the proposed method?**
>
> Response: We apologise for the lack of clarity in our presentation and will strive to improve this aspect, based on your suggestions and questions. What we wished to convey was the following structure:
>
> 1. Prompts in our proposal comprise of a combination of approaches (that provide technical insight) and personas (to bootstrap the model and align it)
> 2. We use the prompting template shown in Figure 6 to request the LLM to generate the names of various approaches. We then construct a word cloud from these approaches using a hold-out set and select the top five. For personas, we directly ask the model to provide a list of individuals suitable for task resolution.
> 3. To generate the few-shot examples, once the approaches are selected (based on frequency of occurrence), we first start with a chain-of-thought demonstration used in Wei et al. (2022), and query GPT-4 to style transfer this using the selected approach, to create an “augmented demonstration”. To ensure the quality of these demonstrations, we conduct a manual evaluation. Given that the task primarily involves a style transfer of the demonstrations, we found no errors. Thus, we have a prompt with (a) a persona, (b) a set of diverse approaches, and (c) corresponding augmented samples.
>
> We agree with the observation made by the reviewer that in practice, our proposal requires a hold out set for evaluating the choice of approaches and personas. However, this set turns out to be very small, with 100 samples (though we believe 50 might often suffice). Additionally, most prompting strategies that are deployed require such a hold out set, and this is not a limitation to our work alone (https://docs.anthropic.com/claude/docs/optimizing-your-prompt).
>
> We also thank the reviewer for other formatting changes suggested to improve the clarity of the paper; we will actively work on making these changes.
>
>
>
> **Merging multiple inference calls into one was first invented and discussed in Batch Prompting ([1] Cheng, et al. Batch Prompting: Efficient Inference with Large Language Model APIs. EMNLP 2023.). Please cite it and tone down your contribution on this technique.**
>
> Response: We thank the reviewer for the pointer to the paper, and will cite it. We would, however, like to highlight the difference between our approach and Batch Prompting. In our approach, we batch multiple “reasoning paths” within the same prompt. Batch Prompting does not (a) specifically address reasoning problems, and (b) primarily involves grouping the response to **“multiple queries together” whereas our approach involves soliciting the response to “one query” using multiple reasoning paths** within the prompt.
>
>
> **Section 2.2. “It independently formulates new approaches” -> Is it a hallucination or a feature? It looks like a hallucination to me. If this is important for achieving good performance, can you provide an ablation study based on whether to allow new approaches or not?**
>
> Response: Since there is no well-defined manner to restrict the model to only following the approaches we propose, we believe it would be hard to perform the ablation. However, from our evaluation of the approaches created (or hallucinated as suggested by the reviewer), the methodology proposed and calculations performed are often accurate. This begs the question as to understanding why these approaches must be restricted. Could the reviewer shed some additional clarity on their thoughts?

---

> > ### Author Response · Authors · 2023-11-20
> > **Any other concerns?**
> >
> > Dear Reviewer,
> >
> > Thanks again for your review! As we near the end of the discussion period, we wanted to ask if we addressed questions you raised, and hope you find our responses useful. We would love to engage with you further if there are any remaining points.
> >
> > We understand that the discussion period is short, and we sincerely appreciate your time and help!

---

> > > ### Author Response · Authors · 2023-11-22
> > > **Thanks for your feedback!**
> > >
> > > Dear reviewer,
> > >
> > > Thanks for your feedback. We have modified the paper based on some suggestions you have provided. We'd appreciate it if you could confirm if the changes meet your requirements, and our rebuttal from earlier is able to satisfactorily respond to your concerns. Today is the last day of the rebuttal period, and we are thankful for your feedback that has helped improve the paper.

---

### Official Review · Reviewer_PAGk · 2023-11-03

**Soundness:** 2 fair
**Presentation:** 3 good
**Contribution:** 2 fair
**Rating:** 3
**Confidence:** 4

**Summary:**

This work proposes DIV-SE and IDIV-SE as prompting strategies to increase diversity of LLM generated responses to reasoning tasks. Instead of relying on the decoding algorithm (sampling) to generate diverse solutions, this approach creates prompts using different approaches from a pool of approaches suitable for a task. When prompted with these different approaches the LLM is shown to generate diverse solutions. Aggregating the results from these diverse prompts has been shown to be more effective than the self-consistency baseline.

The method can be summarised as follows:
1. Create Persona, Approach combinations for a given task: Given a dataset - pick a random question q; ask LLM to come up with Personas & Approaches using LLM output (aggregating m x n suggestions from LLMs based in instruction|question|template ) & then picking pick top 5 using word cloud. Similarly let the LLM generate suggestions for apt personas for a given task.
2. Pick an optimal P,A: For all P,A combinations the composite prompt is tried on a held-out set. And those with the highest performance are selected.
3. Create Augmented demonstrations: Once the approaches for a task are selected as per above, we ask the LLM to modify existing demonstrations with the given set of approaches.
4. Inference: Run with multiple prompts, each with a different augmented demonstration. Then output of these is aggregated. This can be done in a single call to the LLM (IDIV-SE) instead of independent inference steps (DIV-SE).

**Strengths:**

- Clever inference cost optimisation on top of self-consistency which requires making multiple calls to the model for a given task. With the IDIV-SE approach a single call is sufficient to extract different approaches and their corresponding solutions are obtained from the LLM which are then aggregated.
- A novel take on prompting methods, diversity in prompt hasn't been explored to the best of my knowledge.
- Interesting ablation study on error propagation.
- Paper is very well written, most technical aspects are clearly described.

**Weaknesses:**

- The method requires a held out set where different personas and approaches can be tested to pick useful ones. This makes the proposed DIV-SE method somewhat limited in practical applications. CoT prompting on the other hand provide universally applicable prompts, and self-consistency provides a generally applicable framework without needing a held-out set to fine-tune on.

- Results seem mixed: Blocksworld clearly benefits from the diversity in prompting. It is clear on this benchmark that with IDIV-SE more can be achieved in less cost. But on other benchmarks this is not necessarily true e.g. GSM8K (Fig 3 & Fig 4).

- No qualitaitve/quantitative assessment of the synthetically generated demonstrations (using LLMs). For augmented demonstrations - how is the quality of this dataset ensured? I suspect some human annotation to improve the quality of these demonstrations can improve eventual model performance.

- Limited evaluation: Some commonly used reasoning benchmarks are not present, making it hard to compare with prior work like Wang et al's Self Consistency baseline. Arithmetic reasoning: AddSub MultiArith ASDiv SVAMP. Commonsense and symbolic reasoning:  CSQA StrategyQA ARC-e ARC-c Letter Coinflip

- Fig 4 & 5 - For a fair comparison with DIVSE-1 to 5, shouldn't Self Consistency baseline also be given multiple reasoning paths instead of just report SC-1?

- The study could also be extended to other models with smaller sizes e.g. Llama 2 7B variants.
- Difference in effectiveness of DIV-SE could be reported for pre-trained vs chat finetuned/RLHF-ed models.

**Questions:**

- What does it mean to have a zero shot setting of DIV-SE? Isn't it always atleast one shot as shown in Fig 2? I couldn't find details of how this is done in zero-shot setting. Is the prompt appended with list of approaches but no examples?

- What is the quality of these augmentations? Are they really diverse? Are each of them valid? If yes, are they different from other prompts?

- Sec 2.2 . In practice, for all (persona, approach) combinations, we evaluate the prompt formed using the composition on a held-out set and choose those with the highest performance. This seems like a limitation - for open ended reasoning tasks that could appear in conversational applications, how can this be done outside the benchmarks studied?

- Fig 6 & 8 are referenced in the text but missing in the paper? Did you mean Fig 2?

- Can you explain how different this approach is from self-consistency: I'm not sure if the approach can be described as one bringing diversity in prompting. The diversity is being attempted in the set of approaches taken by the LLM to solve a problem. This is still very similar in spirit to self-consistency - where instead of naively sampling n times, the model is prompted to generate different approaches. The approaches themselves are model generated.

---

> ### Author Response · Authors · 2023-11-18
> **Thank you for your feedback!**
>
> [part 1] We thank the reviewer for finding our work clever, novel, and emphasizing the importance of diversity in prompting. We now respond to any concerns the reviewer may have. Should the reviewer have any clarifications, we are happy to further engage. If the responses are satisfactory, we kindly request the reviewer to increase their score towards acceptance.
>
> **The method requires a held out set where different personas and approaches can be tested to pick useful ones. This makes the proposed DIV-SE method somewhat limited in practical applications.**
>
> Response: We agree with the observation made by the reviewer that in practice, our proposal requires a hold out set for evaluating the choice of approaches and personas. However, this set turns out to be very small, often with 100 samples (though we believe fewer, e.g., 50 might also suffice). Prompt engineering is typically a one-time effort, and the resulting prompt could potentially serve thousands or even millions of user queries
>
> In addition, all prompt engineering practices in general (and not only our proposed approach) **do** require a hold out set, but the main issue with most practices in the status quo is that they require a significant amount of trial and error (https://docs.anthropic.com/claude/docs/optimizing-your-prompt).
>
> Additionally, the approaches discovered by our DIVERSE-PROMPTING strategy are general and re-usable across reasoning problems of the same domain, adding to the practicality of our approach. Thus, the cost of evaluating the approaches on a hold out set is further amortised. As an example, for the GSM8K task, some of the LLM-generated approaches include: using visualisations, working backwards, using direct calculation, and method of elimination; these can be reused for other arithmetic reasoning problems.
>
> **Results seem mixed: Blocksworld clearly benefits from the diversity in prompting. It is clear on this benchmark that with IDIV-SE more can be achieved in less cost. But on other benchmarks this is not necessarily true e.g. GSM8K (Fig 3 & Fig 4).**
>
> Response: We would like to stress that the minor improvements are in scenarios where the models are already very performant in the tasks; regardless - the performance using our method is consistently high (with an average improvement of 8 percentage points) and often state-of-the-art (as in the Blocksworld case, with a 29.69 percentage point improvement). All this, with a fraction of the cost of contemporary approaches. We request the reviewer to look at the table (in the summary link below) to learn more about performance gains, and Figure 1 + 3 in the paper to learn more about the cost vs. performance trade-offs.
>
> The reasons for little improvements for some models and datasets is because of task saturation (particularly in the case of GSM8K and GPT-4). This can be for a variety of reasons, such as these tasks not needing substantial reasoning capabilities as the models become bigger, or dataset contamination. However, for tasks such as Blocksworld, reasoning is more important and this is where our method shines compared to prior approaches.
>
> Summary of the results can be found here: https://ibb.co/68PF2Dr
>
> **No qualitaitve/quantitative assessment of the synthetically generated demonstrations (using LLMs).  I suspect some human annotation to improve the quality of these demonstrations can improve eventual model performance.**
>
> Response: In generating the demonstrations for a specific approach, we do not solely rely on the LLM to create the demonstrations from scratch. Instead, we utilize the same chain-of-thought demonstrations from Wei et al. (2022) as a base, and instruct GPT-4 to perform a “style transfer” of the solution using the specific approach.
>
> To ensure the quality of these demonstrations, we conducted a manual evaluation as well. Given that the task primarily involves a style transfer of the demonstrations, we found no errors in the style transferred demonstrations i.e., correctness was preserved.
>
> We also agree with the reviewer that more curation of the “augmented examples” can boost performance. However, we would like to point out that despite the lack of such curation, our approaches demonstrate SOTA performance on tasks that require complex reasoning (e.g., Blocksworld).

---

> > ### Author Response · Authors · 2023-11-18
> > **response part 2**
> >
> > [part 2]
> > **Limited evaluation**
> >
> > Response: Our strategy in selecting the benchmarks was twofold. Firstly, we aimed to choose a diverse set of tasks that required complex reasoning. To this end, we selected tasks involving arithmetic reasoning, algebraic reasoning, commonsense, and planning. Secondly, within each reasoning task, we sought to choose the most challenging benchmark. As shown in Wang et al., 2023, Table 2: AQuA, GSM8K, and CSQA are among the most challenging benchmarks in their respective categories. Similarly, as demonstrated in Valmeekam et al., 2023, the planning benchmark is one of the most challenging for LLMs to solve.
> >
> > **Fig 4 & 5 - For a fair comparison with DIVSE-1 to 5, shouldn't Self Consistency baseline also be given multiple reasoning paths instead of just report SC-1?**
> >
> > Response: The purpose of Figures 4 and 5 is not to compare the performance of these methods directly, but rather to illustrate how the performance of our proposed techniques varies with the ensemble size. These figures provide a comprehensive comparison, showing the performance of the self-consistency baseline and our proposed techniques under the same conditions. Direct comparison between the self-consistency baseline and our proposed techniques is shown in Figures 1 and 3. Here, self-consistency is given as many as 40 reasoning paths (in AQuA), and we note that our approach provides more utility at a lower cost.
> >
> > **The study could also be extended to other models with smaller sizes e.g. Llama 2 7B variants + Difference in effectiveness of DIV-SE could be reported for pre-trained vs chat finetuned/RLHF-ed models.**
> >
> > Response: When evaluating our approach on smaller models (e.g. LLAMA-2 7B), especially if the model is not instruction fine-tuned, its ability to stick to the specification and instructions provided (in generating responses) is very poor. Parsing the results would require significant manual annotation. Additionally, techniques such as chain-of-thought prompt and those that promote multi-step reasoning are most beneficial at models of larger size (https://arxiv.org/pdf/2206.07682.pdf), which is why we also study larger models.
> >
> > **What does it mean to have a zero shot setting of DIV-SE? Isn't it always at least one shot as shown in Fig 2? I couldn't find details of how this is done in zero-shot setting. Is the prompt appended with list of approaches but no examples?**
> >
> > Response: In the zero-shot setting of DIV-SE, we indeed provide no examples. Instead, we only append the name of the approach to the prompt. For instance, the prompt might be appended with “Approach: Using algebra”. Figure 2 illustrates the few-shot setting. In all our experiments, we consistently use the same few-shot examples as those used in the study by Wang at el., 2023.

---

> > > ### Author Response · Authors · 2023-11-18
> > > **response part 3**
> > >
> > > [part 3]
> > > **What is the quality of these augmentations? Are they really diverse? Are each of them valid? If yes, are they different from other prompts?**
> > >
> > > Response: We have manually evaluated all augmented demonstrations to ensure their quality and found no errors. Regarding diversity, as illustrated in Figure 2, the methodology used in the prompts is indeed diverse.
> > >
> > > When self-consistency is run (for the prompt in figure 2), the responses are below:
> > >
> > > Run 1: The candle burns for 5:00 PM - 1:00 PM = 4 hours. Since it melts by 2 centimeters every hour, it will be 4 * 2 = <<4*2=8>>8 centimeters shorter after burning from 1:00 PM to 5:00 PM.
> > >
> > > Run 2: The candle burns for 5:00 PM - 1:00 PM = 4 hours. It melts by 2 centimeters every hour, so after 4 hours, it will be 4 * 2 = 8 centimeters shorter.
> > >
> > > Run 3: The candle burns for 5:00 PM - 1:00 PM = 4 hours. Since it melts by 2 centimeters every hour, the candle will be 4 * 2 = 8 centimeters shorter.
> > >
> > > This clearly highlights the lack of diversity in the approaches.
> > > Can the reviewer suggest a method that we can utilize to quantitatively measure diversity?
> > >
> > > **Fig 6 & 8 are referenced in the text but missing in the paper? Did you mean Fig 2?**
> > >
> > > Response: Figure 6 and 8 are presented in the supplementary material. We will modify the text accordingly.
> > >
> > > **Can you explain how different this approach is from self-consistency**
> > >
> > > Response: The key difference between the two approaches is the nature of the solution paths. Using terminology from our work, the self consistency paper always uses the “same approach” but relies on breaking the problem into smaller subproblems within the approach. However, our work proposes using “different approaches” altogether to solve the problem. This is further highlighted by the fact that the augmentations used in our work are diverse. Additionally, our approach can easily incorporate self consistency within each of the approaches we propose, but the inverse is not true. Most importantly, for a fixed inference budget, our proposed technique outperforms the self consistency method. This indicates that introducing diversity at the conceptual level through various approaches and personas results in more diverse solutions than simply sampling multiple reasoning paths at decoding. **In many ways, diversity-of-thought increases the opportunities for the model to be accurate by purposefully guiding it to use approaches that are relevant to the problem and different from each other.**

---

> > > > ### Author Response · Authors · 2023-11-20
> > > > **Any other concerns?**
> > > >
> > > > Dear Reviewer,
> > > >
> > > > Thanks again for your review! As we near the end of the discussion period, we wanted to ask if we addressed questions you raised, and hope you find our responses useful. We would love to engage with you further if there are any remaining points.
> > > >
> > > > We understand that the discussion period is short, and we sincerely appreciate your time and help!

---

> > > > > ### Comment · Reviewer_PAGk · 2023-11-22
> > > > >
> > > > > Dear Authors, Thank you for your response to many of the concerns and clarifications. I'm however unable to recommend this work for acceptance. The mixed results on the benchmarks studied and the absence of some relevant benchmarks still remain my top concerns here.

---

> > > > > > ### Author Response · Authors · 2023-11-22
> > > > > > **Request for clarifications?**
> > > > > >
> > > > > > Thank you for your feedback, and we apologize if we are unable to satisfactorily resolve your concerns. We are curious on what we can do better -- to this end, we have several questions:
> > > > > >
> > > > > > 1. Are we expecting additional insights from these missing datasets? As we mentioned in our rebuttal, we already pick the ones that prior works perform **least well in**, and substantially perform better than published papers (just last year).  We would like to stress that these experiments are compute intensive, and we'd personally like to avoid running them unless there's a strong reason to, which we hope the reviewer can share. We'd also appreciate information on which "relevant" baseline we have missed.
> > > > > >
> > > > > > 2. With regards to the mixed performance, we'd like to re-iterate that (a) GPT-3.5 (and higher) models are already very performant on a subset of these tasks (e.g., GSM8K) and expecting higher performance (where we already beat previously reported state-of-the art or SOTA numbers) seems strange, and (b) for those datasets where models (including the ones we evaluate) are poor, our technique reports SOTA performance (most often). It is only natural that the performance gains are disparate, but we'd like to stress that the gains are unilateral and often result in SOTA performance. Having demonstrated SOTA performance across most datasets we consider, at substantially lower cost than previously published methods, we find it surprising that the reviewer is recommending against acceptance.

---

> > > > > > > ### Comment · Reviewer_PAGk · 2023-11-23
> > > > > > >
> > > > > > > As I understand the goal of this paper is to design a general prompting strategy that improves the reasoning abilities of LLMs. To be able to find this claim convincing, I'd expect evidence of the method being useful on a few more reasoning tasks beyond Blockworlds and AQUA-RAT. I understand the authors argue that on benchmarks like GSM8K and Commonsense QA, GPT-3.5 like models are performant enough. One alternative could then be proving that DIV-SE and IDIV-SE help attain significant performance gain when applied with models less capable or much smaller than GPT-3.5 on common reasoning benchmarks, or considering a broader set of models besides GPT-3.5 and 4 (similar to the Table 2 from Wang et al 2023 that you pointed out). The 'additional insight' I'm expecting is substantial proof that the method you have proposed is general and effective enough beyond the 2 datasets and the 2 models where DIV-SE & IDIV-SE do provide convincing value.

---

### Author Response · Authors · 2023-11-18
**General Response**

**Concerns re: novelty**

Comparison to Batch Prompting: In our approach, we batch multiple “reasoning paths” within the same prompt. Batch Prompting does not (a) specifically address reasoning problems, and (b) primarily involves grouping the response to **“multiple queries together” whereas our approach involves soliciting the response to “one query” using multiple reasoning paths** within the prompt.


Comparison to Li et al.: In the paper by Li et al., the reasoning problem’s prompt is formed by providing multiple, yet different examples depending on the test sample (refer to figure 10 in their paper), and using a step-wise verifier to correct errors made. **This requires them to have a base of examples from which they can sample few-shot examples for each prompt, a critical limitation as noted in Appendix B.1 of their paper. We do not.**

The “diversity” in their prompts are reasoning paths based on self-consistency for examples from the base of examples (i.e., a hold out set); these reasoning paths are not truly diverse as they all use the same approach (as highlighted in response to reviewer PAGk) and only serve as “more examples” for the model or at best as linguistic diversity triggers (but not reasoning diversity). In contrast, our work (i) involves truly diverse (refer to response to reviewer PAGk) and “related” approaches to solve the reasoning problem (refer to Figure 2 in our work), whilst (ii) limiting the number of examples in the prompt (we use 5, Li et al. need close to 20) and (iii) do not have a step-wise verifier (which further increases the cost).

**Evaluation and Disparate performance**

Selecting Benchmarks: Our strategy in selecting the benchmarks was twofold. Firstly, we aimed to choose a diverse set of tasks that required complex reasoning. To this end, we selected tasks involving arithmetic reasoning, algebraic reasoning, commonsense, and planning. Secondly, within each reasoning task, we sought to choose the most challenging benchmark. As shown in Wang et al., 2023, Table 2: AQuA, GSM8K, and CSQA are among the most challenging benchmarks in their respective categories. Similarly, as demonstrated in Valmeekam et al., 2023, the planning benchmark is one of the most challenging for LLMs to solve.

Difference in Performance: We would like to stress that the minor improvements are in scenarios where the models are already very performant in the tasks; regardless - the performance using our method is consistently high (with an average improvement of 8 percentage points) and often state-of-the-art (as in the Blocksworld case, with a 29.69 percentage point improvement). All this, with a fraction of the cost of contemporary approaches. We request the reviewers to look at Figure 1 + 3 in the paper to learn more about the cost vs. performance trade-offs.

The reasons for little improvements for some models and datasets is because of task saturation (particularly in the case of GSM8K and GPT-4). This can be for a variety of reasons, such as these tasks not needing substantial reasoning capabilities as the models become bigger, or dataset contamination. However, for tasks such as Blocksworld, reasoning is more important and this is where our method shines compared to prior approaches.

**Need for a hold-out set**

We agree with our proposal requires a hold out set for evaluating the choice of approaches and personas. However, this set turns out to be very small, often with 100 samples (though we believe fewer, e.g., 50 might also suffice). Prompt engineering is typically a one-time effort, and the resulting prompt could potentially serve thousands or even millions of user queries

In addition, all prompt engineering practices in general (and not only our proposed approach) **do** require a hold out set, but the main issue with most practices in the status quo is that they require a significant amount of trial and error (https://docs.anthropic.com/claude/docs/optimizing-your-prompt).

---

### Meta-Review · Area_Chair_nXro · 2023-12-11

**Metareview:**

This paper develops a diversity of thought prompting method to encourage the diversity of the reasoning process and then ensemble the results together. Different from previous ensemble method that that create diversity of thought through decoding process, it works on improving the diversity of thoughts by including multiple solution paths in the prompt. However, there are still several major weaknesses, such as mixed experimental results, lack of more comprehensive benchmark evaluations, clarification in presentations, etc. The authors are encouraged to further improve the paper accordingly and submit to a future venue.

**Justification For Why Not Higher Score:**

There are still several major weaknesses in the current version. The paper still need significant further work in order to be accepted.

**Justification For Why Not Lower Score:**

N/A

---

### Decision · Program_Chairs · 2024-01-16

Reject